# Localisation of Ancient Migration Pathways inside a Fractured Metamorphic Hydrocarbon Reservoir in South-East Hungary

**Tivadar M. Tóth [1,\*], László Molnár [1,2], Sándor Körmös [1], Nóra Czirbus [1] and Félix Schubert [1]**

[1] Department of Mineralogy, Geochemistry and Petrology, University of Szeged, 6720 Szeged, Hungary; molnar.laszlo@rhk.hu (L.M.); sandor.kormos@geo.u-szeged.hu (S.K.); czirbus.nora@geo.u-szeged.hu (N.C.); schubert@geo.u-szeged.hu (F.S.)

[2] Public Limited Company for Radioactive Waste Management (PURAM), 2040 Budapest, Hungary

\* Correspondence: mtoth@geo.u-szeged.hu

**Abstract:** Numerous fractured hydrocarbon reservoirs exist in the metamorphic basement of the Pannonian Basin in Hungary. Many decades of experience in production have proven that these reservoirs are highly compartmentalised, resulting in a complex mosaic of permeable and impermeable domains situated next to each other. Consequently, in most fields, only a small amount of the total hydrocarbon reserve can be extracted. This paper aims to locate the potential migration pathways inside the most productive basement reservoir of the Pannonian Basin, using a multiscale approach. To achieve this, evaluation well-log data, DFN modelling and a composition analysis of fluid trapped in a vein-filling zeolite phase are combined. Data on a single well are presented as an example. The results of the three approaches indicate the presence of two highly fractured intervals separated by a barely fractured amphibolite. The two zones are probably part of the communicating fracture system inside the single metamorphic mass. The gas analysis further specifies the migrated fluids and indicates hydrocarbons of a composition similar to that of the recently produced oil. Consequently, we conclude that the two zones do not only form an ancient migration pathway but are also members of a more recent hydrocarbon system.

**Keywords:** Pannonian Basin; fractured reservoir; well-log; DFN modelling; fluid inclusion chemostratigraphy

---

## 1. Introduction

Recently, the role of naturally fractured reservoirs has become increasingly important in numerous fields of applied geology. In this unique type of reservoirs, the complex network of fractures inside a rock body acts exclusively as a space for the migration and storage of lithospheric fluids, such as water and hydrocarbons [1]. Whether or not such a fracture network can transport fluids depends on several factors determined by the tectonic and post-tectonic evolution of the area in question. First, the hydrodynamic behaviour of the network is determined by how the individual fractures are spatially organised (e.g., [2–5]). The most essential geometric parameters related to this pattern include the density of the fracture system and the orientation and size of the individual fractures. In numerous cases, structural evolution results in mutual connectivity throughout the fracture network, while in other cases, a compartmentalised network may develop, with the fracture system containing sub-systems that do not communicate with each other [6–8]. In addition to geometrical influences, other factors may also reliably modify the behaviour of the fracture system. For example, certain fractures may close due to increasing lithostatic pressure (depth; [4]). In addition, recent anisotropic stress fields have had a significant effect on the opening and closure of fractures with different orientations [4]. In addition to these factors, the conditions of the fractured

rock body can be modified by various water–rock interaction processes, such as the dissolution of the host rock and the precipitation of new mineral phases, leading to the formation of mineralised veins.

In Hungary, the most productive hydrocarbon reservoirs can be found in the Neogene basin-filling sedimentary sequences of the Pannonian Basin [9–11]. Excellent fractured reservoirs also exist in the pre-Neogene basement, being related to Mesozoic carbonates [12] and Variscan metamorphic rock bodies (e.g., [13]). The hydrodynamic behaviour of these reservoirs differs significantly from that exhibited by typical porous systems. In most cases, impermeable blocks of rocks are separated by highly fractured zones with good reservoir conditions, while elsewhere, rock bodies themselves may contain mutually connected fracture systems. In some localities, the productive intervals communicate with each other, resulting in an entirely connected hydrodynamic system. In addition, there are other basement hydrocarbon fields in the Pannonian Basin in which hydrocarbons of various chemical compositions are produced, suggesting the co-existence of several reservoirs that do not communicate with each other hydrodynamically [14]. In the most extreme situation, 12 wells produce seven types of immiscible oils in a small area on the order of 10 km$^2$, indicating the existence of a highly compartmentalised fractured basement reservoir [15].

In this study, we focus on one of the most productive basement hydrocarbon fields in the Pannonian Basin. Over 100 wells penetrate the metamorphic basement in the Szeghalom Dome, most of which produce hydrocarbons [16–19]. Nevertheless, until now, less than 10% of the total estimated amount has been produced, essentially because of the rather complicated internal structure of the reservoir. The present paper aims to locate possible paleo-migration pathways and the most productive intervals along a single well in the Szeghalom Dome by using a combined approach involving well-log data evaluation, fracture network modelling, and ancient fluid composition analysis.

## 2. Geological Setting

In the Szeghalom area, the basement of the Pannonian Basin comprises Variscan metamorphic rocks (Figure 1; [16]). A detailed petrological analysis has revealed that three blocks with varying metamorphic evolutions can be distinguished, which are separated by wide (~10 to 20 m) post-metamorphic structural boundaries [19]. Through evaluations of seismic profiles, a complex system of low- and high-angle fault zones has become evident [20]. The deepest structural level includes a block dominated by homogeneous orthogneiss (OG; [21]), above which lies a polymetamorphic garnet- and sillimanite-bearing paragneiss block (SG block). The topmost block is dominated by amphibolite and amphibole gneiss (AG; Figure 2). Molnár et al. [19] evaluated composite well-logs in the area and distinguished the above lithologies from the rocks of the tectonised major fault zones. Through further refinement of the well-log evaluation algorithm, the most essential tectonic rock types, such as cataclasite, fault breccia and fault gauge, can be classified. Based on an interpretation of the well-logs from numerous distinct wellbores, the spatial extension of the fault zones between the SG and AG blocks can be determined to be at a low angle (8°–13°) on top of the N–NW thrust sheets between them cut by younger normal faults, which is in harmony with seismic interpretations (Figure 2). The structural boundary between the OG and SG blocks exhibits similar characteristics, but its presence is less evident.

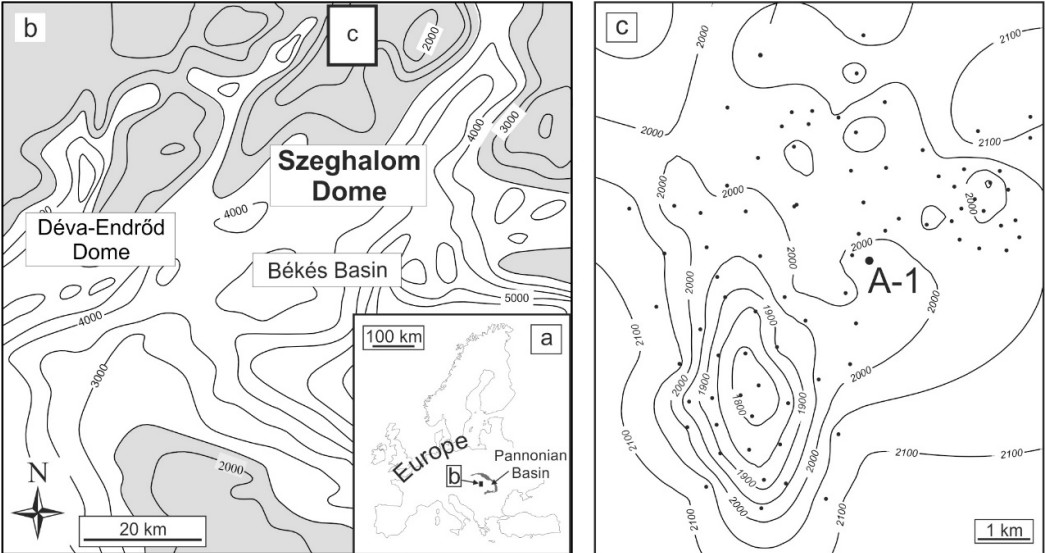

**Figure 1.** Sketch of the study area. (**a**) Position of the Pannonian Basin in Europe. (**b**) Subsurface topographic map of the SE part of the basement of the Pannonian Basin. Isolines denote depth (in m) below the present surface (m). (**c**) Topographic map of the Szeghalom metamorphic dome. Dots represent wells reaching the basement of the Pannonian Basin.

The fracture intensity and geometric parameters of the fracture network substantially differ for the three lithological realms. By far, the most intense brittle deformation is typical in the topmost AG block. Previous models have suggested a communicating fracture system for the massive amphibolite bodies, while the fracture networks of both the schistose SG and gneissic OG blocks usually appear to be disconnected [17].

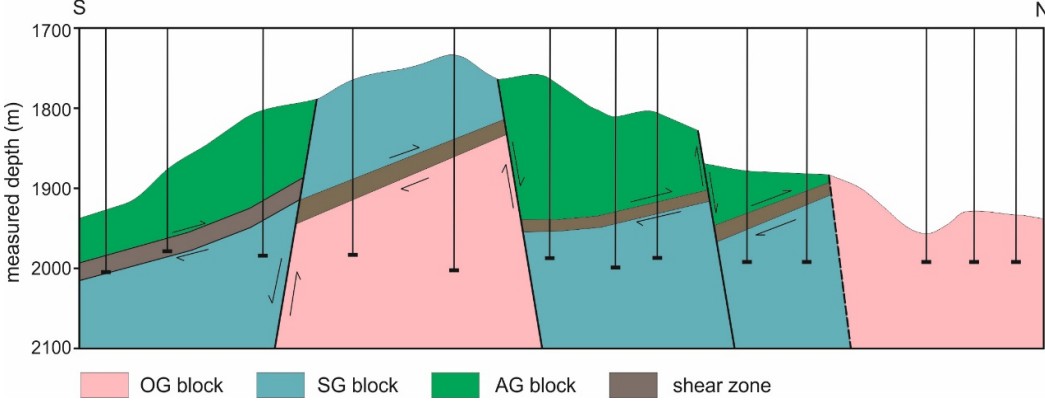

**Figure 2.** N–S geological section across the highest point of the Szeghalom metamorphic dome (modified from [19]).

Nevertheless, the vein-filling minerals are identical throughout the area, independent of the host rock, and are characterised by the following sequence: pyrite → chlorite/kaolinite → calcite1 → quartz → calcite2 → laumontite (Figure 3). Juhász et al. [22] discovered that the first five phases precipitated during a period of continuous cooling in which the basement uplifted to the surface. This was later confirmed by M. Tóth et al. [23], who described plant fragments and pollen inclusions in the calcite2 phase, representing a Miocene terrestrial flora. Schubert et al. [18] also analysed hydrocarbon-bearing fluid inclusions in vein-filling quartz crystals and determined that the formation temperatures ranged between 120 °C and 135 °C (Figure 3). Moreover, they found oil inclusions with various compositions in different segments of the reservoir, indicating its temporally and/or spatially compartmentalised behaviour when these ancient hydrocarbons migrated into the

fracture system. The final phase in the veins was laumontite, which is assumed to have formed under recent reservoir conditions [22].

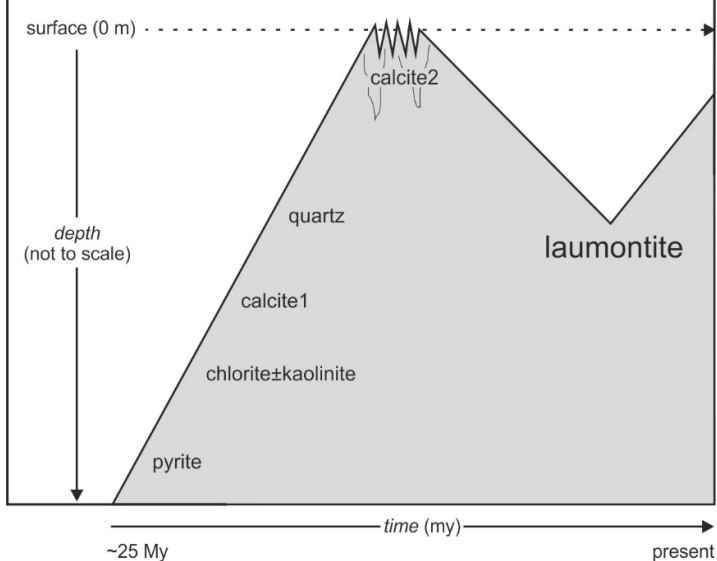

**Figure 3.** Exhumation and subsidence history of the study area revealed by Juhász et al. [22] and Schubert et al. [18].

The well that was studied in this project (A-1) was drilled during the 1980s in the northern part of the area (Figure 1c) and penetrated the metamorphic basement with ~100% core recovery rate between 2050 and 2080 m below the present surface. The rock material of the total bore core, as well as the above set of well-logs, is available for complex evaluation.

## 3. Methods

The conditions of the fractured reservoir were analysed using well-logs, rock specimens and vein-filling laumontite associated with the A-1 well. To characterise the penetrated interval lithologically, the standard petrographic microscopy analysis was applied.

All available well logs (gamma ray—*GR*, resistivity—*Res*, density—*Den*, and compensated neutron—*CN*) were evaluated using Molnár et al.'s [19] method, to localise the tectonised zones and distinguish different tectonised rock types in the Szeghalom Dome. Based on the several depth intervals for which both rock specimens and well-log data were available, Molnár et al. [19] derived a linear function using discriminant function analysis to discriminate the unaltered host rock from the tectonites. The optimal function was found to be *F(1-2) = 1.1 \* GR − 0.5 \* Res − 0.9 \* Den*, whose values were significantly lower for undeformed rock bodies than those typical in shear zones (Figure 4a). To further discriminate the three basic tectonic types of the Szeghalom area, two functions are required, with the followings being derived for this purpose: *F3 = 0.7 \* CN − 0.5 \* Res* and *F4 = 0.6 \* Den − 0.9 \* GR*. Using these two discriminating variables, cataclasites, fault breccias, and fault gouges can be used to define independent clouds on an appropriate plot (Figure 4b). In the present study, these functions were applied to the case of A-1 well, and the results were compared with those previously obtained for the neighbouring wells.

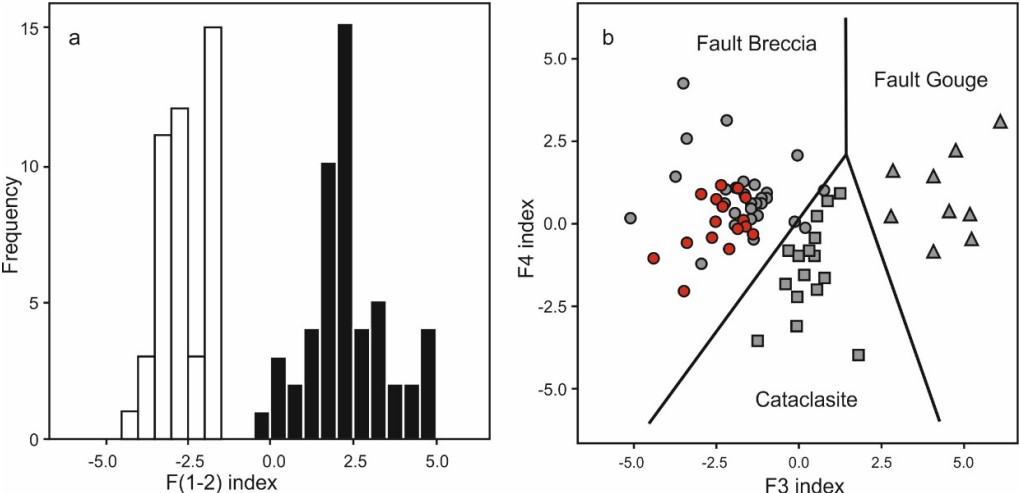

**Figure 4.** Discriminant functions to distinguish (**a**) undeformed host rock (white bars) from fault rocks (black bars) and (**b**) fault breccia, cataclasite and gouge based on the discriminant functions *F(1-2)*, *F3* and *F4* calculated using Discriminant Function Analysis on well-log data from the Szeghalom area (modified after [19]). Red dots denote the fault rocks of the studied A-1 well.

To simulate the fracture network, the typical geometric parameters were first measured along the well, using high-resolution core images. In the first step, the position of each fracture was recorded along a reference line, resulting in a series of intersection points. This point series was evaluated to derive three fracture density parameters: the spacing (inter-point distance), P10 index and fractal dimension of spacing.

The spacing log exhibits the variation in distances between any two neighbouring fracture intersections along a well, and thus visualises the varying density in segments of the fracture network. The frequently used parameter P10 [24] measures the number of fractures along any 1-m-long interval. In contrast to the spacing, P10 increases in more intensely fractured intervals. For the simulation, a third parameter is worth deriving. As fracture networks are usually handled as 3D fractal-type geometries [25,26], the intersection of the network with any line results in a fractal pattern embedded in one dimension [27]. There are numerous methods for computing the fractal dimension of such a point series, such as box counting, roughness length analysis, and special variogram analysis [28]. M. Tóth [28] introduced a multi-step approach for using the rescaled range (R/S) analysis to calculate the fractal dimension (*D*) of any segment of a fracture network crosscut by a bore core. Using this approach, a *D*-log can also be constructed along the well and compared with the two other fracture density logs. In this paper the M. Tóth's [29] algorithm to 6-m-long overlapping intervals was applied. In this way, the *D*-value can be calculated every 3 m.

The fracture length data of a fracture system in metamorphic rocks usually follow a power-law distribution through numerous scales, ranging from centimetres to tens of metres [30,31]. That is, N(*L*) = F * $L^{-E}$, where *L* is the trace length of a fracture on any 2D section, N is the number of fractures of length *L* and E (the length exponent) and F are constants. By determining the trace length of several individual fractures, their distribution function can be computed, and thus, E and F can be computed. To avoid unpredictable bias in this estimation, only those fractures that terminated inside the bore core were used in the image analysis. Because only a few hundred individual fractures were available for evaluation, a unique pair of parameters (E, F) could be calculated for the entire well, while the spatial variation of the length distribution could not be followed. As the studied A-1 well was not oriented, no primary information concerning the real geographical position of the bore cores was available. For this reason, the fracture orientation data, dip, and dip angle measured by a borehole televiewer (BHTV) in a neighbouring well were used for the simulation.

Considering all the above information, in the fracture network simulation, the fracture density was computed as a fractal dimension (*D*) for each 3-m-long interval, fracture lengths were

characterised by a unique (E, F) pair for the whole well and fracture orientation data, which should have been statistically identical for the entire rock body, were applied during the simulation.

For fracture network modelling, we used the newly developed Infress software, which applies the modified algorithms of the previous RepSim code [17]. This is a fractal geometry-based discrete fracture network (DFN) modelling system, in which each fracture is represented by a circle. The simulated 3D network had the same statistical parameters as the original natural network concerning spatial density (fractal dimension of fracture midpoints) as well as fracture size (diameter), distribution, and orientation (dip and dip angle). Moreover, the simulation was a stochastic process, meaning that any number of different models of identical probability could be generated using the same initial set of parameters. In the present case, 10 near-well fracture models were produced around the well and evaluated for a 30 × 30 m² column. As a result, interconnected fracture sub-systems as well as their spatial positions were determined for all 10 models. Further details on the mathematical background of the simulation procedure have been discussed in many previous papers [32–35].

Considering the vein-filling mineral phases, all, except for the final laumontite, represent ancient mineralisation processes, as shown previously [22]. Thus, hereinafter, only the conditions of laumontite formation will be focused on. In the first step, zeolites separated from five distinct depths were measured by X-ray diffraction to check whether other zeolite phases occurred in the veins. Zeolite separates were handpicked, powdered, and analysed by a Philips PW1710 automated diffractometer using Cu-Ka radiation at the University of Bern (Switzerland). The chemical compositions of laumontite crystals from three different depths were analysed using a Cameca SX-50 electron microprobe at the University of Bern. Synthetic and natural standards were used, and the conditions included a 15 kV electric potential and a 20 nA beam current. Natural minerals [36] were used as standards. To model the stability conditions (pressure, temperature, and fluid composition) of the vein-filling laumontite, the THERIAK-DOMINO thermodynamic system was applied [37,38], which calculates equilibrium mineral paragenesis for any $P_0$-$T_0$ point. When modelling fluid-bearing systems, an appropriate $T$-$X_i$ ($P$-$X_i$) field should be modelled instead. In this study, the $T$-$X_{CO2}$ field (at constant pressure) will be applied. As input, the bulk chemical composition of the vein-filling mineral paragenesis and an internally consistent thermodynamic database are required. In this project, the Berman [39] system, completed with data of some Ca-zeolites (laumontite, heulandite, stilbite, and wairakite), was used.

To facilitate the understanding of the paleo-accumulation and migration of hydrocarbons, as evidenced previously by Schubert et al. [18], fluid inclusion chemostratigraphy (FiCs) was employed using a single-mineral-phase laumontite (as the last vein-filling cement phase) along with bulk sidewall rock samples. A FiCs analysis involves mass spectrometry without chromatographic pre-separation. The procedure is based on a technology called fluid inclusion stratigraphy or fluid inclusion volatile [40]. The method aims to analyse the organic and inorganic volatile compounds trapped in fluid inclusions or exclusions from drill cuttings, core samples or outcrop samples. However, here, the fluids tightly held in nanopores and other confined pores/spaces were also analysed and contributed to the results. Due to the nature of these healed vacuoles, the captured liquids provided essential information concerning the ancient fluid system.

Laumontite was separated from sidewall rock samples under a stereomicroscope. Considering the adsorption capacity of the laumontite [41], neither inorganic nor organic solvents were used during the sample preparation. Samples weighing 1 g were placed in a high-vacuum–pneumatic crushing chamber at 100 °C and $10^{-5}$ mbar for 10 min. Then, the vacuum-cleaned samples were crushed, and the liberated volatile compounds were electron ionised and swept into a quadrupole PFEIFFER PrismaPlus™ QMG 220 mass spectrometer. The molecular species were analysed using an electron multiplier in the scanning mode according to their mass-to-charge ratio (m/z) in the range of 2–100. The acquisition time was set to 500 ms/amu.

In total, 21 pieces of laumontite and 38 pieces of bulk sidewall rock were selected for the FiCs analysis. In addition to the FiCs depth logs, the obtained mass spectra were also evaluated independently. According to the literature [42–45], non-hydrocarbon and hydrocarbon species were

chosen for normalisation to compare the recorded ion currents among the samples. Moreover, the acquired species were normalised against the total response of each sample. Furthermore, to make the signals comparable among individual samples, the total response was normalised against the maximum reading of each sample set.

## 4. Results

### 4.1. Petrography and Mineralogy

Based on both macroscopic and microscopic analyses, the penetrated interval was found to consist of amphibolite or amphibole gneiss (Figure 5a). Meanwhile, each studied amphibolite sample comprised hornblende + plagioclase + ilmenite ± quartz. While the amphibolite samples were massive, the few-tens-of-centimetre-thick amphibole gneiss members were foliated and mainly comprised hornblende + quartz + K-feldspar + plagioclase + biotite + ilmenite. The entire penetrated interval was found to be intensely fractured (Figure 5a), with fault breccia layers of several centimetre width at depths of ~2050, 2062, and 2066 m. Furthermore, the results indicated that one brecciated section between 2071 and 2076 m reached 5 m in thickness.

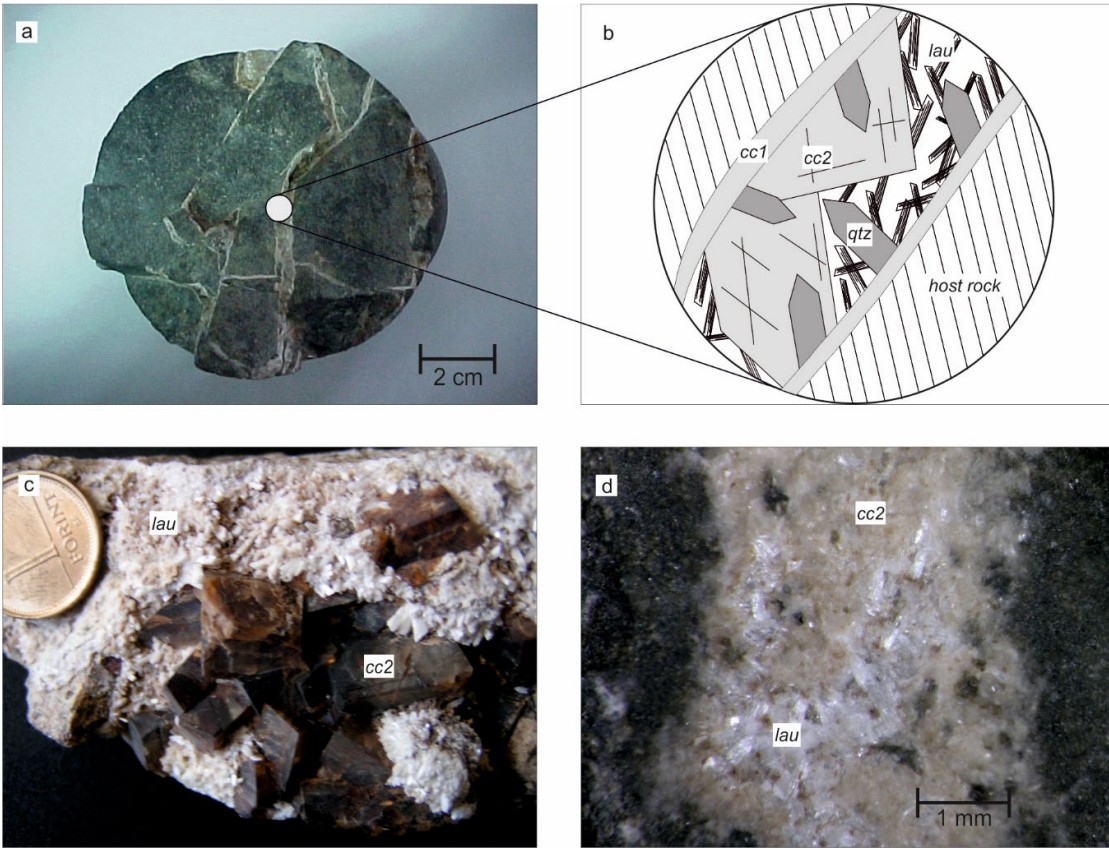

**Figure 5.** Typical vein-filling mineralogy of the amphibolite. (**a**) Representative fractured amphibolite bore core. (**b**) Sketch of the vein-filling mineral sequence. (**c**) Open vein with brown calcite2 and laumontite. The coin is 16 mm in diameter. (**d**) Vein filled by calcite2 and laumontite.

The ideal sequence of the vein-filling minerals is chlorite/kaolinite → calcite1 → quartz → calcite2 → laumontite (Figures 5b,c), which is typical throughout the region. Numerous entirely mineralised veins were also found to contain only chlorite and calcite, while the closing zeolite phase appeared exclusively in open fractures and brecciated zones (Figures 5c,d). X-ray diffractometry indicates that laumontite is the only zeolite mineral of the vein-filling sequence. It has a close-to-ideal composition with Na < 0.4 and K < 0.1 pfu (formula calculated for 48 O; [46]. The two alkalis revealed

a tight positive correlation, while Na + K increased coevally with the loss of Ca, indicating a Na + K ⇌ Ca substitution (Figure 6).

To understand the conditions under which potential laumontite-bearing parageneses might be formed, a mineral assemblage that sufficiently mimics the vein composition was chosen. In particular, the behaviour of the theoretical 10 calcite + 6 kaolinite + 3 quartz mineral composition was modelled. We used the THERIAK-DOMINO software for modelling in the $T$-$X_{CO_2}$ field [37,38], with a temperature of 125 °C < $T$ < 225 °C and a fixed pressure of P = 400 bars. The $CO_2$ concentration changed from 0 to being oversaturated. The results indicated that zeolite could not appear at high (>0.5) $X_{CO_2}$ fluid compositions. Stilbite and heulandite appeared under moderate $X_{CO_2}$ levels, while wairakite was stable at $T$ > 210 °C. The narrow field in which laumontite became stable was at a low temperature (<200 °C) and extremely low $X_{CO_2}$ levels (<0.15; Figure 7). The reaction *calcite + kaolinite + quartz + H₂O = laumontite + CO₂* may be responsible for laumontite formation in the veins.

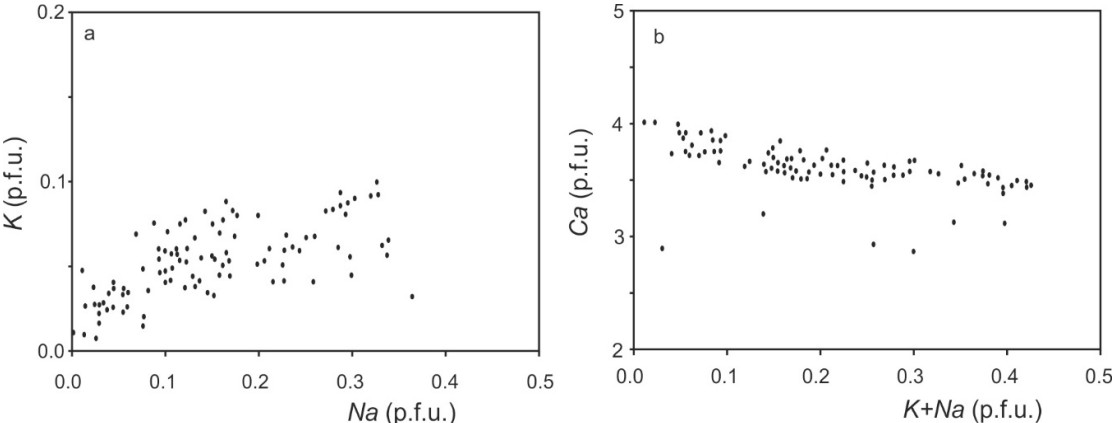

**Figure 6.** Chemical compositions of single laumontite crystals. (**a**) Na vs. K. (**b**) K + Na vs. Ca. Formula calculated based on 48 O.

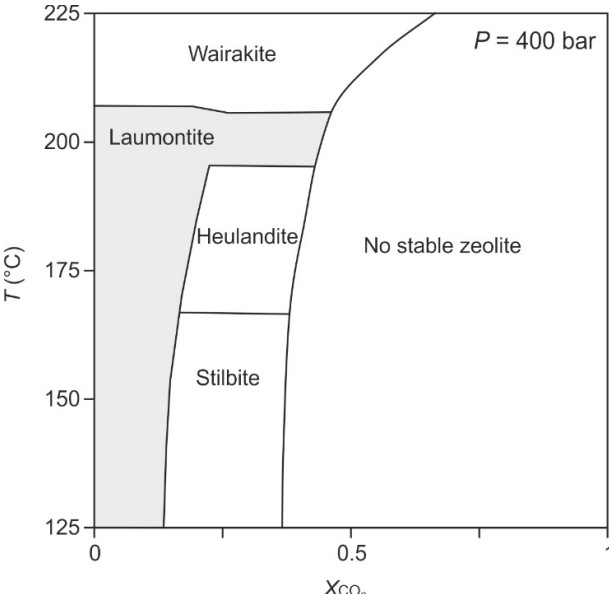

**Figure 7.** Results of the laumontite stability calculations on $T$-$X_{CO_2}$ field (THERIAK-DOMINO modelling software; [38].

### 4.2. Well-log Evaluation

Applying the discriminant functions for the well-logs, two thick (~5 m) tectonised layers could be localised within the homogeneous AG block: one at a depth of 2051–2057 m and the other at 2073–2079 m. Additionally, a thinner, less characteristic layer was observed at the depth interval between 2062 and 2065 m (Figure 8). Based on the values of functions *F3* and *F4*, the internal structure of the thick tectonised zones was dominated by coarse fault breccias (Figure 4b).

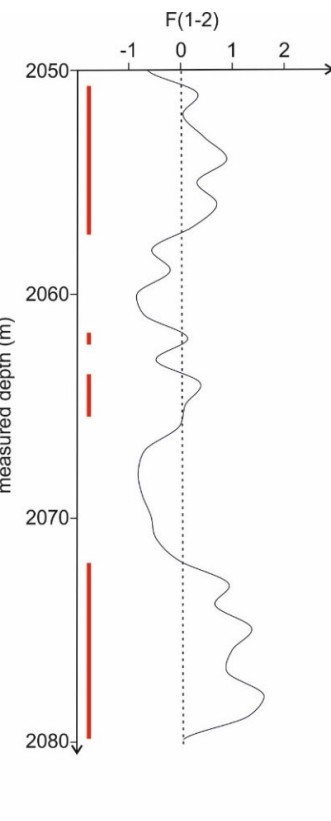

**Figure 8.** Vertical variation in the *F(1-2)* values along the A-1 well. The dashed line separates the undeformed host rock from the faulty rocks (Figure 4a). Red bars indicate the possibly deformed intervals.

### 4.3. Fracture Network Modelling

An image analysis of the core photographs revealed ~650 individual fractures. This dataset was evaluated to obtain reliable geometric parameters of the area along the well. The spatial distribution of the fractures indicated that the entire 30-m-long section was intensively deformed, with varying fractured intervals. In certain locations, the neighbouring fractures were only a few centimetres away from each other, while the largest spacing was ~40 cm (Figure 9a). The variation in the P10 values also indicated a slightly heterogeneous fracture density undergoing alterations between 16 and 30 m$^{-1}$. This index, along with P10-log, revealed three depth intervals with particularly large values (>25 m$^{-1}$): ~2052, 2060, and 2070 m (Figure 9b). As the fractal dimension (*D*) accumulates information for 6-m-long sections, *D*-log exhibited an even smoother curve. Its values varied between 1.07 and 1.50, again with three extremes at depths of ~2052, 2065, and 2075 m. While the spacing and P10 had no meaning inside the ~5-m-long brecciated interval at ~2075 m, the *D*-value could be computed using the neighbouring data (Figure 9c).

As numerous fractures were found to not terminate inside the bore core, only ~550 fractures yielded reliable data length. Their measured values varied between 3 and 25 cm with a modus of 9.5 cm. The tight correlation (r = 0.96) between log*L* and logN(*L*) revealed that the data followed a power-

law distribution. The length exponent was computed as the slope of the regression line on this plot, resulting in E = −4.78 (Figure 10).

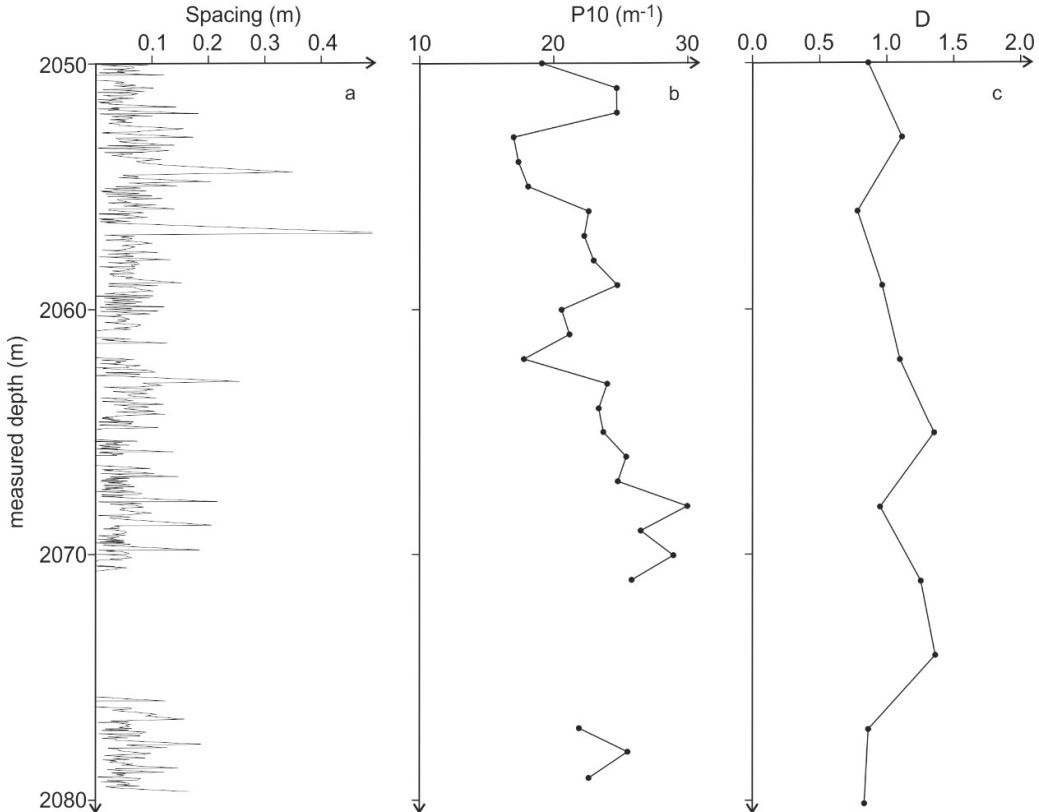

**Figure 9.** Fracture density logs along the A-1 well. (**a**) Spacing log. (**b**) P10 parameter log. (**c**) Fractal dimension (*D*) log. For definitions of the parameters, please refer to the text.

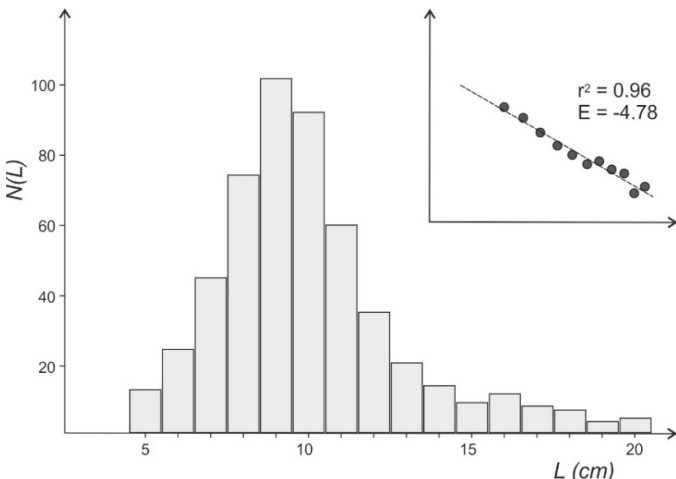

**Figure 10.** Fracture length distribution measured along the A-1 well. Inset: the log*L* − logN(*L*) plot of the histogram for estimating the length exponent E.

All 10 independent near-well models showed similar results, indicating two intensively fractured zones at ~2065 and 2075 m, where individual fractures formed a communicating network (Figure 11). Some simulations indicated that these zones were interconnected, while others exhibited a compartmentalised network of two entirely isolated sub-systems (Figure 11). Based on the

consistent result of all models, the zone at ~2052 m comprised individual fractures, which probably do not define a mutually interconnected network.

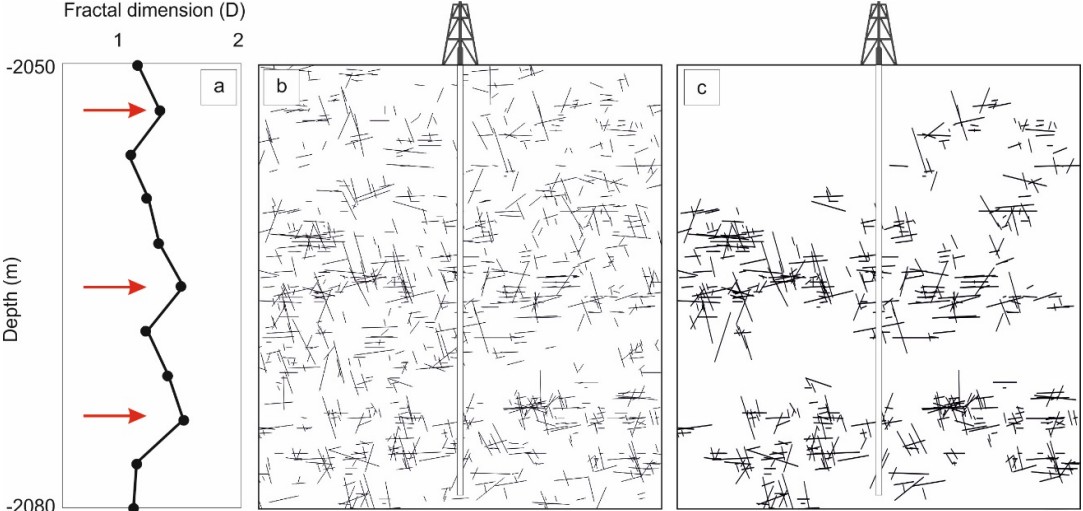

**Figure 11.** Representative fracture network model. (**a**) Fractal dimension (*D*) log. Red arrows denote the intervals with high fracture density. (**b**) Representative vertical section of the total fracture network around the A-1 well. (**c**) Communicating part of the same fracture network.

### 4.4. Fluid Composition

The results of the FiCs analyses were jointly examined due to the common features of the fluid remnants trapped in the laumontite and sidewall rock samples. The obtained mass spectra revealed the presence of non-hydrocarbon and hydrocarbon species within the liberated volatiles (Figure 12). The compound groups formed clusters on the spectra, enabling the recognition of individual components and/or molecular fragments present in certain compounds. Predominantly, two different fluid compositions characterised the mass spectra of the fluid residues along the investigated core section (Figure 12), with the volatile non-hydrocarbon species being generally dominated by $H_2O$ and $CO_2$ (Figures 12a,b). In the investigated mass range, the hydrocarbon fraction contained compounds with a maximum of seven carbon atoms and with a dominance of $CH_3^+$ fragments, with maximum part representing methane. In addition, aromatic hydrocarbon species were detected between the described clusters, highlighting the presence of benzene and toluene in the trapped oils (Figure 12c). The fluid remnants exclusively characterised by non-hydrocarbon or hydrocarbon species complied with two extreme cases; however, their mixture was more common in the remained fluids, demonstrating the co-occurrence of the volatile species (Figure 12b).

In both sample sets, the normalised contributions of hydrocarbon and non-hydrocarbon species were found to vary, especially in the sidewall rock samples. The variation had a fluctuating pattern, indicating an abundance of methane and water (Figure 13).

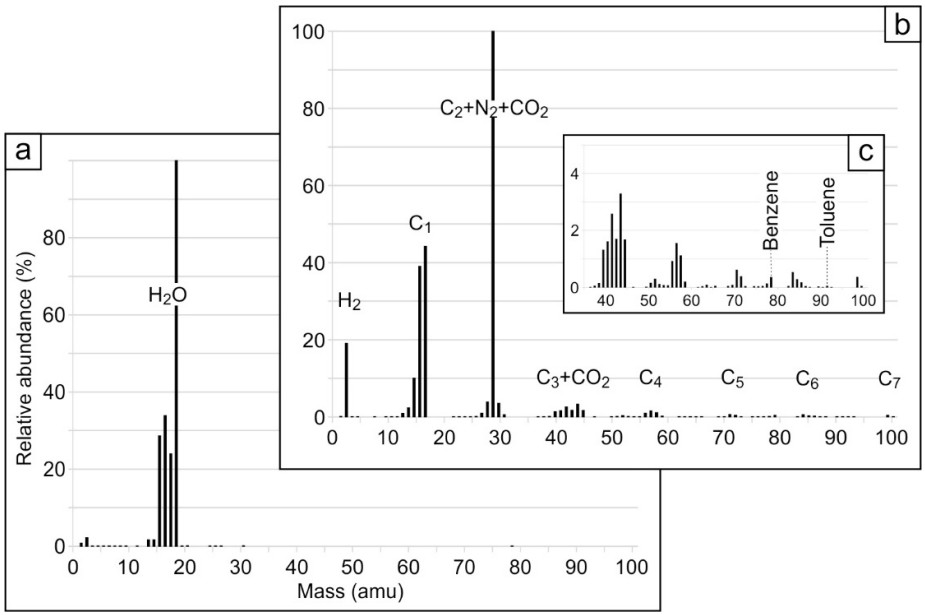

**Figure 12.** Mass spectra of the fluid remnants characterising (**a**) non-hydrocarbon and (**b**,**c**) hydrocarbon species.

There was also a definite appearance of $H_2O$ isolate intervals with a dominance of aqueous fluids. Within these regions, benzene, together with higher-carbon-numbered hydrocarbons, also occurred, though hydrocarbons existed independently too, indicating their predominance. Focusing on the laumontite samples, the hydrocarbon-bearing fluids clearly established a dominant phase, with a minor contribution from aqueous fluids. The maximum readings of benzene were a close match between the bulk rock and laumontite samples between 2061 and 2067 m, as well as 2072 and 2077 m, but the declared sections were assigned a maximum difference of 1 m (Figure 13).

## 5. Discussion

### 5.1. Characterisation of the Fracture Network

According to the well-log interpretation, the total interval of the metamorphic basement penetrated by the A-1 well is lithologically homogeneous. Petrographic data confirm that the well crosscut is exclusively amphibolite and amphibole gneiss, which are typical rock types of the AG block. Although the presence of the SG block can be assumed below this block [16,19], the shear zone that separates the two lithological realms throughout the study area is not reached by the well.

Nevertheless, the well-log interpretation indicates three depth zones, with the host rock being intensively deformed at each zone. Two of these (~2052 and ~2075 m) were also dominated by brecciated amphibolite (Figure 8). The parameter weights in the discriminant function *F(1-2)* define the main petrophysical differences between the undeformed host rock and the tectonised zones. The most significant factor in the separation is the high natural gamma-ray value, which indicates an elevation in the amount of clay minerals caused by the intensive weathering and alteration of the host rock along the fault zone. At the same time, the relatively low electrical resistivity of the tectonites reflects considerable fluid infiltration into these zones. The lower densities probably relate to the higher frequency of fractures in the tectonised zones, coupled with porosity enhancement.

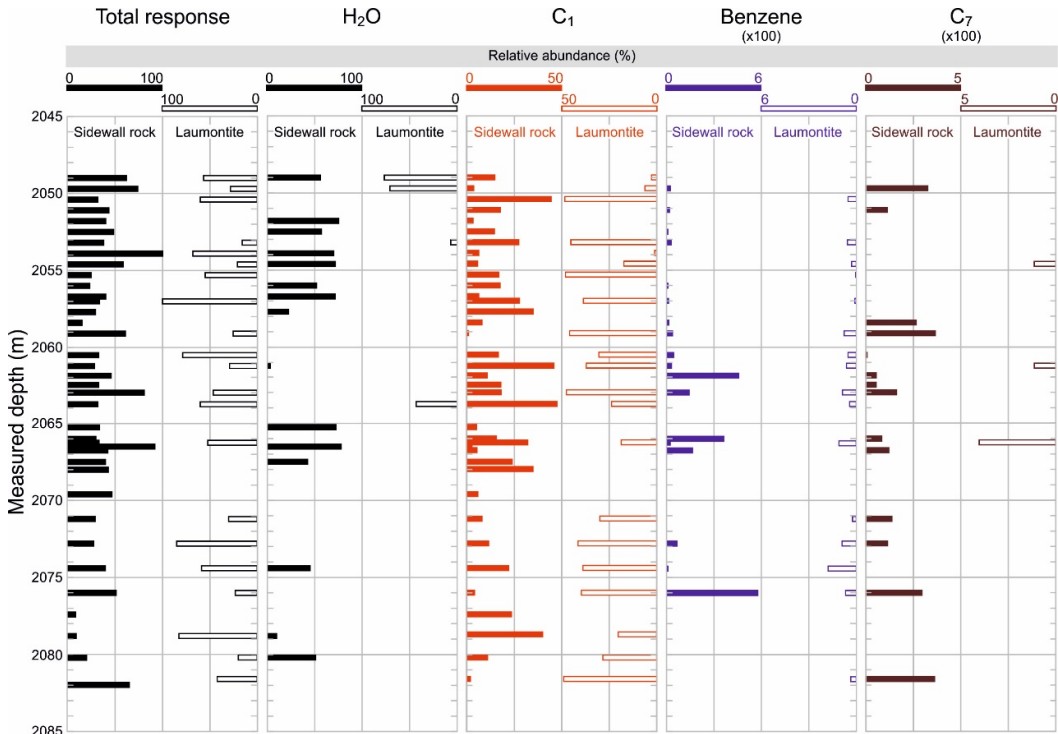

**Figure 13.** Fluid inclusion chemostratigraphic log of the A-1 well.

According to the function *F3*, higher levels of deformation are manifested inside the brecciated layers due to an increase in the compensated neutron porosity values and a further decrease in electrical resistivity. In the function *F4*, higher density and lower gamma-ray values are characteristic. These petrophysical features indicate that the intensely deformed fault cores (cataclasites and fault gouges) can be characterised by low densities and resistivities, along with elevated natural gamma activity and compensated neutron porosity values. This is consistent with the observations of both [47] and [48].

In case of well A-1, the applications of *F3* and *F4* functions indicate that the two thick deformed intervals at ~2052 and 2075 m depths are mainly composed of coarse fault breccias. Furthermore, based on the *F3* and *F4* discriminant functions (Figure 4b), these zones are characterised by higher density and resistivity values, along with lower gamma-ray and neutron-porosity scores within the fault rocks. These values suggest that weaker disaggregated zones can maintain a significantly higher porosity than comminated fault cores. Previous studies on the structure of fault zones [49,50] have clarified that similar fault breccia layers are typically related to a weakly disaggregated, densely fractured "damage zone", which is generally associated with higher conductivity and permeability relative to the undeformed protolith or the clay-rich fault core.

The thin deformed layer between 2062 and 2065 m has lower *F(1-2)* values, indicating that compared to the brecciated zones at ~2055 and 2075 m, this zone is affected by weaker deformation not intense enough to trigger fault breccia formation. Nevertheless, this process is presumably so pervasive that it forms an interconnected fracture network, also able to act as a migration pathway.

The brittle deformed character of all these zones is also demonstrated by the core material. The zone at a depth of ~2075 m is a thick (~5 m) fault breccia zone, while the layer at ~2065 m is not brecciated but falls into numerous thin layers separated by an intensively fractured host rock. The behaviour of the zone at ~2052 m is more contradictory. Based on the well-log evaluation, the zone appears as a brecciated layer, while the cores indicate an intensively fractured host rock. A comparison of this well-log pattern (Figure 8) with those identified in the neighbouring wells (not presented) reveals a similar appearance in all wells North of A-1. These comparison wells exclusively penetrate the rocks of the AG block and reveal the sheared zones inside the tens-of-metre-thick

amphibolite sequence. Nevertheless, the wells to the South penetrated only a massive amphibolite. As previous studies [16,19] have revealed, the presence of an SG block can be inferred below the AG block, separated by wide (~10–20 m) fault zones throughout the Szeghalom Dome. Such an arrangement suggests that the shear zones inside the AG block can be spatially extended and have a south-dipping position parallel to the major thrust sheet that separates it from the underlying SG block. Wells both to the East and West also have rocks of SG and OG blocks below the AG block. In harmony with the structural evolution of the area [16,19,20], these regions represent elevated horsts separated by normal faults from an AG-dominated block between them (Figure 14). Based on their position and lithological characteristics, the brecciated amphibolite layers in the A-1 well and its neighbouring wells can be assumed to relate to the overthrust zone between the SG and AG blocks at smaller scales. From this viewpoint, the structures revealed by the A-1 well can be interpreted as minor fault zones accompanying the larger fault zone within the hanging wall (i.e., AG block).

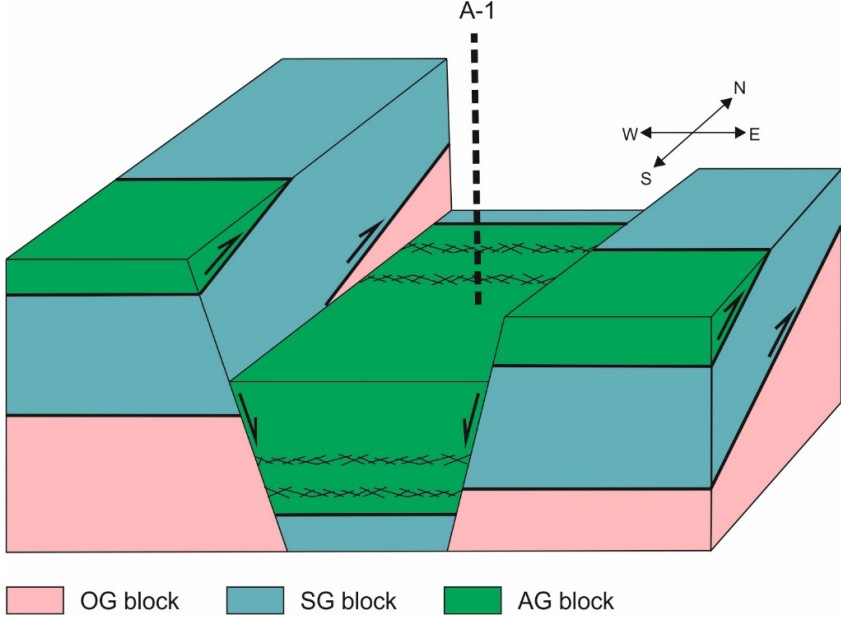

**Figure 14.** Block model representing the structural position of the A-1 well and its surroundings. The two highly fractured zones inside the AG block with good reservoir properties are denoted.

The near-well fracture network models based on the geometric parameters of the fractures measured along with the A-1 well indicate communicating fracture systems in two of the three well-deformed depth intervals (~2075 and 2065 m; Figure 11). In contrast, the zone at ~2052 m most likely contains individual fractures, which do not define a connected network. At these points, the output of all 10 independent simulations reveals fracture patterns with identical behaviour, a result sufficiently stable to be accepted. Nevertheless, the connectivity of the fracture system outside the two well-developed layers is more ambiguous, with some models exhibiting communication between the two highly fractured zones and others indicating two entirely isolated fracture systems (Figure 11). The reason behind such unpredictable behaviour is that outside the two brecciated zones, the fracture network is probably geometrically close to the percolation threshold [51], meaning that the communication between the two highly fractured zones is limited [52].

Combining the results of the well-log interpretation and fracture network modelling, three intensively tectonised intervals can be outlined inside the AG body. Because of the mutual interconnection of the fracture network resulted by the DFN models (Figure 11), two of them probably exhibit substantial lateral extension. Whether or not they communicate hydrodynamically can hardly be predicted using the fracture system simulation approach.

## 5.2. Formation of Laumontite

Previous studies have demonstrated that vein-filling mineralogy is identical throughout the study area [22]. These investigations also ascertain many details regarding the formation of the early phases of the mineral sequence. Based on the reconstructed formation temperatures, the early chlorite/kaolinite, calcite1, quartz and calcite2 records represent the subsequent steps of exhumation of the metamorphic massif to the surface. Plant fragments enclosed in calcite2 crystals represent a middle Miocene terrestrial flora, indicating that by this time, the metamorphic rocks had reached the surface and probably formed islands in the Pannonian Sea [23], Figure 3). This scenario is further confirmed by the Badenian age of the conglomerate covering the uplifted metamorphic dome [10].

The results of experimental petrology indicate that the minimal stability temperature of laumontite is above ~125 °C. As this zeolite phase follows calcite2 in the vein-filling sequence, it has to crystallise during or following the subsidence phase of the basin's evolution. The facts that the veins lack all other zeolite minerals and that the composition of laumontite is rather stable along the well also indicate identical formation conditions throughout the rock body. The reasons for the slight variation in the chemical composition of laumontite are not known in detail. Kiseleva et al. [53] found that while the Ca-Na exchange is an endothermic reaction, the Ca–K exchange is exothermic. Thus, the coeval $Ca \rightleftarrows (Na + K)$ cation exchange (Figure 6) is probably caused by the changing composition of the fluid that occurs during the formation. A similar fluid-dependent mineral composition was identified in the case of other zeolites, such as in the heulandite characterised by Fridriksson et al. [54].

In the present case, laumontite could have formed as a product of the reaction of the previous vein-filling phases according to the reaction *calcite + kaolinite + quartz + $H_2O$ = laumontite + $CO_2$*, as shown above. Based on the classification scheme developed by Greenwood [55,56] and Kerrick [57], this reaction is a mixed $H_2O$-$CO_2$ type, where the two fluid components occur on opposite sides of the chemical equation [58]. In agreement with the THERIAK-DOMINO model shown above (Figure 7), extremely low $X_{CO2}$ levels are required to stabilise laumontite in such a system. As the reaction produces $CO_2$ continuously, its concentration can be sufficiently low only if the deliberated gas can migrate away, i.e., in the case of an open fracture system.

As presented above, the amount of laumontite is found to increase significantly around the two highly fractured intervals, while closed veins, far from the communicating brecciated zones, are missing zeolite infill. Thus, based on the formation requirements, laumontite with extremely low $X_{CO2}$ levels is an excellent indicator of communicating fracture systems, such as in the two modelled depth zones in the A-1 well.

## 5.3. Consequences for Migration Pathways

In this study, FiCs analysis is applied to enhance the understanding of the migrated and accumulated fluids trapped within the investigated samples. Based on Juhász et al. [22], and Schubert et al. [18], a mixture of different fluid regimes (e.g., magmatic, metamorphic, meteoric, and hydrothermal) along with the corresponding volatile compositions can be assumed when interpreting the FiCs results. Comparing the sidewall rock and laumontite samples, both differences and similarities are evident in the pattern of individual mass spectra and chemostratigraphic logs based on the core. The responses of the selected non-hydrocarbon and hydrocarbon species are generally sufficient, making interpretation possible.

The obtained scattering patterns of the $H_2O$ and $CH_3^+$ compounds in the sidewall rock samples indicate the presence of multiple fluid inclusion assemblages and/or migrating fluids with different compositions. Laumontite, as the final vein-filling phase, is possibly less affected by fluid migration events, indicated by the fact that a smooth and even log with a substantial hydrocarbon signal is recorded (Figure 13). Despite the abovementioned distribution in the sidewall rock samples, the cogenetic behaviour of the immiscible fluids cannot be excluded due to the co-occurrence of water, methane and water-soluble hydrocarbons (e.g., benzene and toluene), as has already been discussed previously (e.g., [59,60]).

In both sample sets, whenever $C_7$ hydrocarbons are present, an exponentially decreasing pattern characterises the $C_{2+}$ hydrocarbon range (Figure 12b). Based on the characteristic fragmentation of the

hydrocarbons (e.g., [61]), the obtained mass spectra indicate the contribution of higher-carbon-numbered hydrocarbons to the recorded spectra. The aromatic hydrocarbons benzene and toluene are also present in relatively great abundance (Figures 12c and 13). The contribution of benzene to the FiCs signal is almost constant, with some fluctuations in case of the laumontite samples. In contrast, three intervals in the sidewall rock samples are richer in benzene, and these zones overlap with the corresponding maximum loci detected in the laumontite (Figure 13). The assumed composition of hydrocarbons ($C_{7+}$ and aromatics) indicates the presence of gas condensate or liquid hydrocarbons (e.g., light oil; [62]) in the fluid remnants trapped in the samples. This presumption is consistent with the American Petroleum Institute (API) gravities of the oils produced in the research area [63], indicating that the explored zones were not only active during ancient fluid conduction but also part of a more recent hydrocarbon system.

As petrological modelling has demonstrated, laumontite is an excellent indicator of a communicating fracture system. The FiCs analysis indicates that the fracture-filling laumontite preserves a unique stage of fluid migration when hydrocarbons are present throughout the connected fracture network. Laumontite, which encloses hydrocarbons of a composition similar to that produced recently in oil, probably crystallises coevally with hydrocarbon migration. The overlap between the enhanced benzene content of the fluid remnants in the sidewall rock and in the laumontite samples indicates that microcracks exist within the country rock, serving as conduits for hydrocarbon migration. The elevated abundance of the aromatics (e.g., benzene and toluene), liquid hydrocarbons (e.g., $C_7$) and co-existing water suggests migrating hydrocarbons with various water saturations in the depth range between 2061 and 2067 m, as well as between 2072 and 2077 m. However, besides the inferred differences in water saturation of particular fluid migration events, the results might also have been affected by contributions of the remnants of earlier fluid regimes (e.g., magmatic or metamorphic).

In the sidewall rock samples, the abundance of hydrocarbons is found to decrease to 2060 m and a generally water-dominated zone is detected. The low amount of benzene in this zone may be a result of diffusion of the water-soluble fraction and can be used as a proximity indicator of petroleum accumulation [64]. The dispersedly occurring heavier hydrocarbons might have originated from the same hydrocarbon emplacement event, or represent the remnants of another migration process, as described by Schubert et al. [18].

The FiCs analysis of the drill core from the A-1 well provides further evidence of the communicating fracture systems found at the same depth intervals by the other methods, confirming the results of the well-log interpretation and fracture network modelling.

## 6. Conclusions

Multiscale interpretation of the A-1 well in the fractured metamorphic Szeghalom basement reservoir indicates two intensively fractured, hydrodynamically active zones separated by an impermeable amphibolite body. Through a statistical evaluation of gamma ray, resistivity, density, and compensated neutron logs, the undeformed host rock can be distinguished from the rocks of shear zones. On this basis, three intensively deformed zones were localised along the well area. Further refinement of the method indicated that these zones comprised fault breccias rather than cataclasites. The results of fractal geometry based DFN simulation using geometric data from the fractures (length, spatial density, and orientation) indicated an interconnected fracture system in two of the abovementioned deformation zones. These two zones can be spatially extended and are characterised by mutual fracture connectivity, while the third zone is likely below the percolation threshold. The final vein-filling phase in these sheared intervals is laumontite, which, as it is stable exclusively under low $X_{CO2}$ conditions, is a good mineral indicator of open communicating fracture systems. Chemostratigraphic logs of the sidewall rock and laumontite samples revealed various organic and inorganic volatile compounds trapped as fluid remnants. The elevated abundance of aromatics, liquid hydrocarbons and co-existing water indicates the presence of migrating hydrocarbons with various water saturations in the two depth intervals dominated by fault breccia. The assumed composition of these hydrocarbons is similar to those produced recently in oil,

indicating that the explored zones were not only active during the conduction of ancient fluids but are also members of a more recent hydrocarbon system.

**Author Contributions:** Conceptualization, T.M.T. and F.S.; well-log interpretation, L.M.; organic geochemistry, S.K. and N.C. All authors have read and agreed to the published version of the manuscript.

**Funding:**   This research received no external funding

**Acknowledgments:** The authors thank Enago (www.enago.com) for the English language review.

**Conflicts of Interest:** The authors declare no conflict of interest.

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

**Publisher's Note**: MDPI stays neutral with regard to jurisdictional claims in published maps and institutional affiliations.

