# Peer review of "Localisation of Ancient Migration Pathways inside a Fractured Metamorphic Hydrocarbon Reservoir in South-East Hungary"

_applsci, doi:10.3390/app10207321_

Round 1
Reviewer 1 Report
Dear authors,
I found you article very interesting regarding the research topic and the methodology.
The manuscript can be improved, there are some statements without clear basis and figures wit low quality. There are also some weird expressions, and overlapping symbology for variables.
I just made few comments in an attached manuscript.
Good luck

Author Response
- Thanks for the thorough review and the helpful comments.
- Most of the suggestions are accepted. We rephrased all sentences suggested and added all new references.
- We also improved the quality of the figures, redrew them if needed.
- The term „function” is replaced by „index”.
- Abbreviation „D” is used for fractal dimension, while „F” is used for discriminant functions throughout the text.
Reviewer 2 Report
The topic is interesting and the paper contains original ideas, but the manuscript needs some minor revisions. I have attached a file with all my corrections/suggestions essential to improve the paper.

Author Response
- Thanks for the thorough review and the helpful comments.
- Most of the suggestions are accepted.
- We rephrased all sentences suggested and added the new references.
- We also redrew the bad quality figures.
Reviewer 3 Report
An interesting article, but it was possible to explain in a more accessible way the connection of individual paragraphs with the thesis of the article, and to indicate clearly which factors from the analyzes performed with discriminant functions give the best results and which only indicate statistical significance. Is it not worth considering which methods of the performed analyzes were redundant in the performed analyzes?
Minor notes are included in the attached pdf as comments to the content. I suggest extending the list of articles cited further to include authors from outside Hungary.

Author Response
- Thanks for the thorough review and the helpful comments.
- Most of the suggestions are accepted.
- We rephrased the sentences and paragraphs suggested to be clearer. In several cases also new sentences were inserted. The role of Discriminant Function Analysis in discriminating lithologies in the study area is clarified.
- 13 is now referred to in the text before the figure itself.
- As the well-log interpretation of the neighbouring wells is not presented in the manuscript, to locate them in fig. 14 without any relevant geological or geophysical information would not help to understand the paper. So, we do not accept this suggestion.
- We think none of the applied methods redundant. They all have their role in understanding deeper details of the behaviour of the studied reservoir. The main message of the paper is that very different methods should be combined for a better understanding of complicated fractured reservoirs.
- Altogether 44 of the total 64 references represent the international literature (methodology and analogies) and only 20 of them cite the geology of the study area.